# True Few-Shot Learning with Language Models

**Ethan Perez**[1], **Douwe Kiela**[2], **Kyunghyun Cho**[13]
[1]New York University, [2]Facebook AI Research,
[3]CIFAR Fellow in Learning in Machines & Brains
perez@nyu.edu

## Abstract

Pretrained language models (LMs) perform well on many tasks even when learning from a few examples, but prior work uses many held-out examples to tune various aspects of learning, such as hyperparameters, training objectives, and natural language templates ("prompts"). Here, we evaluate the few-shot ability of LMs when such held-out examples are unavailable, a setting we call *true few-shot learning*. We test two model selection criteria, cross-validation and minimum description length, for choosing LM prompts and hyperparameters in the true few-shot setting. On average, both marginally outperform random selection and greatly underperform selection based on held-out examples. Moreover, selection criteria often prefer models that perform significantly worse than randomly-selected ones. We find similar results even when taking into account our uncertainty in a model's true performance during selection, as well as when varying the amount of computation and number of examples used for selection. Overall, our findings suggest that prior work significantly overestimated the true few-shot ability of LMs given the difficulty of few-shot model selection.

## 1 Introduction

Major progress in language model (LM) pretraining has led to the idea that LMs can learn a new task using a small number of examples only, i.e., few-shot learning [1–3]. Few-shot learning overcomes many challenges with data-rich supervised learning: collecting labeled data is expensive, often requires experts, and scales poorly with the number of tasks. However, the few-shot performance of LMs is very sensitive to the textual task description ["prompt"; 3–6], order of training examples [6–8], decoding strategy [9, 10], and other hyperparameters [3, 5, 9, 11, 12], as well as the learning algorithm itself [3, 12]. Thus, effective model selection is crucial for obtaining good few-shot performance.

There are issues with how recent work approaches model selection in few-shot learning, however. Prior work uses large train or held-out sets with many examples to choose prompts [2, 12, 13] and hyperparameters [12]. Other work claims to use no validation set for hyperparameter selection [3, 11, 14] but does not describe how they design other aspects of their learning algorithm (e.g., training objectives). It is unlikely that no validation examples were used, given the sophisticated nature of the proposed algorithms. In this work, we examine if prior few-shot learning methods still perform well when using only the provided examples for model selection, a setting we term *true few-shot learning*.

We find that true few-shot model selection yields prompts that marginally outperform random selection and greatly underperform selection based on held-out examples. Our result shows that prior work may have greatly overestimated the few-shot ability of LMs. In other words, one reason that prompts are so effective ["worth many examples"; 15] is that they are often tuned using many examples. We evaluate two standard model selection criteria – cross-validation (CV) and minimum description length (MDL) – finding that both obtain only limited improvements over random selection and perform much worse than selection using held-out examples. For prompt selection, our observation holds for 9 LMs ranging over 3 orders of magnitude in size [1, 2, 16] on 3 classification tasks and 41

35th Conference on Neural Information Processing Systems (NeurIPS 2021).

tasks in the LAMA benchmark [17]. For choosing hyperparameters, true few-shot selection causes performance to drop by 2-10% across 8 tasks for ADAPET [12], a state-of-the-art few-shot method. Furthermore, true few-shot model selection has high variance in performance; selected models often do much worse than randomly-chosen ones. We find similar results when varying the number of examples used, amount of computation, and conservativeness of our selection criterion. Altogether, our results suggest that model selection is a fundamental roadblock to true few-shot learning.

## 2 Can We Do Model Selection in Few-Shot Learning?

Prior work uses the phrase "few-shot learning" in multiple senses, raising questions about what it means to do few-shot learning. We categorize few-shot learning into three distinct settings, each of which assumes access to different data. Here, we formally disambiguate between these settings to help future work avoid inadvertently comparing few-shot methods that operate in different settings.

Consider the supervised learning scenario where we have a dataset of inputs $x_{1:N}$ and labels $y_{1:N}$, sampled from a distribution over datasets $D$. We aim to determine the learning algorithm $\mathcal{A}^* \in \mathcal{A}_1, \ldots, \mathcal{A}_A$ with the smallest generalization loss $\mathcal{L}$ at predicting $y$ given $x$ on unseen validation examples $D_{\text{val}} \sim D$ after learning on training examples $D_{\text{train}} \sim D$. We say that an algorithm $\mathcal{A}(D_{\text{train}}, R)$ maps a training dataset $D_{\text{train}}$ and various random factors $R$ that influence training to a function that predicts $y$ given $x$. $\mathcal{A}$ specifies, e.g., the model architecture, hyperparameters, and prompt. $R$ includes random factors that impact the results of a learning algorithm, such as parameter initialization and the order of training examples for online learning algorithms like stochastic gradient descent. We say that $\mathcal{A}$ obtains a generalization loss $\mathcal{L}(\mathcal{A}(D_{\text{train}}, R), D_{\text{val}})$ on a given validation set $D_{\text{val}}$. We aim to find the $\mathcal{A}^*$ that minimizes the expected loss across training and validation sets:

$$\text{EL}(\mathcal{A}, R) = \mathbb{E}_{D_{\text{train}}, D_{\text{val}}} \left[ \mathcal{L}\Big( \mathcal{A}(D_{\text{train}}, R); D_{\text{val}} \Big) \right]$$

In *data-rich supervised learning*, $\text{EL}(\mathcal{A}, R)$ is usually evaluated with a single train-validation split $(D_{\text{train}}, D_{\text{val}})$. Since large $D_{\text{train}}$ and $D_{\text{val}}$ are not always available, the traditional few-shot setting evaluates $\text{EL}(\mathcal{A}, R)$ with many small $(D_{\text{train}}, D_{\text{val}})$ drawn from many, distinct distributions $D$ [see, e.g., work in meta-learning 18–21]. Each distribution $D$ is sampled from $D^*$, a distribution over distributions (e.g., of similar tasks), so we call this setting *multi-distribution few-shot learning*.

Recent work does not assume access to data from other distributions, performing few-shot learning using only a few examples from a single distribution to update a pretrained LM [2, 12]. These papers use a large validation set $D_{\text{val}}$ to tune the learning algorithm $\mathcal{A}$, a setting we term *tuned few-shot learning*. For example, Brown et al. [2] try prompts with different phrasings and numbers of training examples to improve the validation accuracy of GPT-3. Tam et al. [12] choose the early stopping iteration, prompt, and other model-specific hyperparameters based on validation performance. Tuned few-shot learning relies on many labeled examples, so we argue that tuned few-shot learning does not qualify as few-shot learning. If many validation examples are available, they could be incorporated into the training set and trained on using data-rich supervised learning. Tuned few-shot learning algorithms should be compared against data-rich supervised learning algorithms that use the same amount of total data $|D_{\text{train}}| + |D_{\text{val}}|$.

In this work, we evaluate the success of tuned few-shot learning methods when no large $D_{\text{val}}$ is available, a setting we term *true few-shot learning*. Formally, we aim to choose a learning algorithm $\mathcal{A}$ with low expected loss $\text{EL}(\mathcal{A}, R)$ using only a small training set $D_{\text{train}}$ drawn from a single distribution. Here, we must choose $\mathcal{A}$ by approximating $\text{EL}(\mathcal{A}, R)$, e.g., using cross-validation. Several papers claim to circumvent the need to estimate $\text{EL}(\mathcal{A}, R)$ by choosing hyperparameters based on an educated guess [3, 9, 14]. However, the proposed learning algorithms themselves are quite sophisticated, and it is unclear how they were designed if not by using validation performance. Other work chooses the learning algorithm and hyperparameters using one or multiple other datasets before evaluating on the target dataset [5, 11]. Such approaches fall under *multi-distribution few-shot learning* and cannot be directly compared to methods that attempt to perform true few-shot learning, even though prior work has made such comparisons [14].

In what follows, we describe two model selection criteria – cross-validation and minimum description length – which we use to evaluate tuned few-shot methods in the true few-shot setting.

## 2.1 Cross-validation

Cross-Validation (CV) [22–24] is one of the most widely used methods for estimating generalization loss [25]. CV has also been used in prior work on multi-distribution few-shot learning [26, 27]. CV randomly partitions $D_{\text{train}}$ into $K$ equally-sized folds $F(D_{\text{train}})_1, \ldots, F(D_{\text{train}})_K$ and evaluates the average loss on a validation fold $F(D_{\text{train}})_k$ when training on the remaining data $F(D_{\text{train}})_{\neg k}$:

$$\text{CV}(\mathcal{A}, R, F) = \mathbb{E}_{k \sim \text{Unif}(1, K)} \left[ \mathcal{L}\Big(\mathcal{A}(F(D_{\text{train}})_{\neg k}, R); F(D_{\text{train}})_k\Big) \right]$$

In this way, CV forms $K$ train-validation splits out of the pool of labeled examples. CV with one example per fold ($K = N$ folds) is commonly referred to as leave-one-out CV (LOOCV).

## 2.2 Minimum description length

We may also form train-validation splits in a different manner than CV, drawing inspiration from work on the Minimum Description Length (MDL) principle [28]. MDL can be estimated by evaluating the average loss on a fold $F(D)_k$ when training on the previous folds $F(D)_{1:k-1}$:

$$\text{MDL}(\mathcal{A}, R, F) = \mathbb{E}_{k \sim \text{Unif}(1, K)} \left[ \mathcal{L}\Big(\mathcal{A}(F(D_{\text{train}})_{1:k-1}, R); F(D_{\text{train}})_k\Big) \right]$$

This procedure is referred to as "online coding" [29, 30], as it evaluates the generalization loss of the algorithm as it learns "online" from more and more data.[1] There are other ways to evaluate MDL [see 31, for an overview]. We use online coding as it has been shown to be an effective way to estimate MDL, especially for deep learning methods [32].

MDL measures generalization because it evaluates how much a learning algorithm compresses the labels $y_{1:N}$ given the inputs $x_{1:N}$, and because better compression implies better generalization [33]. Recent work has used MDL to determine which learning algorithms are most effective at explaining the given data [Rissanen Data Analysis; 10, 34].

## 2.3 Variance matters

We evaluate the generalization loss of the algorithm chosen by CV (likewise for MDL):

$$\mathcal{L}(\mathcal{A}_{\text{CV}}(D_{\text{train}}, R), D_{\text{val}}), \qquad \text{where } \mathcal{A}_{\text{CV}} = \arg\min_{\mathcal{A}} \mathbb{E}_{R,F}[\text{CV}(\mathcal{A}, R, F)].$$

The above loss should be low in expectation, across different datasets $D_{\text{train}} \sim D$, $D_{\text{val}} \sim D$, and random factors $R, F$: $\mathbb{E}_{D_{\text{train}}, D_{\text{val}}, R, F}[\mathcal{L}(\mathcal{A}_{\text{CV}}(D_{\text{train}}, R), D_{\text{val}})]$. The loss should also be low in variance: $\mathbb{V}_{D_{\text{train}}, D_{\text{val}}, R, F}[\mathcal{L}(\mathcal{A}_{\text{CV}}(D_{\text{train}}, R), D_{\text{val}})]$. Low variance implies that CV/MDL *reliably* choose an algorithm that generalizes to $D_{\text{val}}$ when trained with a given $D_{\text{train}}$ and random factors $R, F$. Reliability is important for many practical or commercial applications where worst-case performance is important, such as image recognition [35, 36], dialogue systems [37, 38], and robotics [39, 40].

We also experiment with explicitly taking into account an algorithm's variance during model selection, choosing $\mathcal{A}_{\text{CV}}$ to minimize a conservative estimate of CV, $\text{CV}_\alpha(\mathcal{A})$, chosen such that the probability $\text{Pr}_{R,F}[\text{CV}(\mathcal{A}, R, F) < \text{CV}_\alpha(\mathcal{A})]$ is high:

$$\text{CV}_\alpha(\mathcal{A}) = \mathbb{E}_{R,F}[\text{CV}(\mathcal{A}, R, F)] + \alpha\sqrt{\mathbb{V}_{R,F}[\text{CV}(\mathcal{A}, R, F)]}$$

where $\alpha$ is a hyperparameter set based on the desired probability. In particular, if $\text{CV}(\mathcal{A}, R, F)$ follows a normal distribution $\mathcal{N}$ when sampling $R, F$, then $\text{CV}(\mathcal{A}, R, F) \leq \text{CV}_\alpha(\mathcal{A})$ with probability $\int_{-\infty}^{\alpha} \mathcal{N}(\mu = 0, \sigma = 1)$ for a given $R, F$. $\text{CV}_\alpha(\mathcal{A})$ resembles the Watanabe Akaike Information Criterion [41], which estimates the generalization of a model trained with $\mathcal{A}$ using the expected loss from a model trained with $\mathcal{A}$ plus the variance in training loss across models trained with $\mathcal{A}$.

---

[1]Online coding formally computes a sum over $\mathcal{L}(.)$ rather than the expectation, which differs by a constant factor. The two are equivalent for our purposes (ranking $\mathcal{A}$).

## 2.4 Other model selection criteria

Prior work has developed other model selection criteria such as the Akaike Information Criterion [AIC; 42], Watanabe-Akaike Information Criterion [WAIC; 41], and Mallows' $C_p$ [43]. These methods often rely on assumptions or quantities that are not available in the context of deep learning (AIC, Mallows' $C_p$) or are approximations of LOOCV (WAIC). Since state-of-the-art few-shot learning methods tend to be based on deep learning, we focus on CV and MDL as our model selection criteria. In Appendix §B, we also test several other criteria that are applicable to deep learning methods.

Selection criteria can be optimized automatically, e.g. with bayesian optimization [44–46], evolutionary methods [45, 47, 48], reinforcement learning [49], or gradient descent [50–53]. Such methods aim to match the performance of exhaustive search, the optimal approach (used in our work).

## 3   True Few-Shot Prompt Selection

Recent work on LMs performs few-shot learning by providing training examples as input in the form of a natural language "prompt" [2, 3, 9]. For example, for a question-answering task, Brown et al. [2] prepend input examples with "READING COMPREHENSION ANSWER KEY" before providing them to GPT-3 (see Appendix Table 2 for more examples). They then have the LM complete the remaining words in the prompt, conditioning on earlier words (including various input examples), following the LM's pretraining objective (next word prediction). No parameter updates are involved. It is not obvious *a priori* which prompts will generalize well for a given LM, and there is high variance in how well different prompts generalize [3, 6], even between prompts with minor differences [e.g., one comma; 5]. Effective prompt selection is thus crucial for true few-shot learning.

### 3.1   Experimental setup

In what follows, we test on LAMA [17], a benchmark for retrieving facts with LMs, for which prior work has developed many strategies for designing prompts [4, 54–56]. LAMA evaluates the accuracy of LMs at choosing the correct target object for various (`subject`, `relation`, `object`) triples present in knowledge bases, such as (`Dante`, `born-in`, `Florence`). We use the "TREx" split [57], which consists of 41 relations (up to 1k examples each). Petroni et al. [17] design a prompt for each relation, which an LM completes to predict an answer (e.g., "The birthplace of Dante was _"). Some relations have multiple valid target entities, so LAMA evaluates how often one of the true answers matches the top-predicted token (out of 20k candidates). We only use examples from the LAMA-UnHelpfulNames subset [LAMA-UHN; 58] which filters out easy-to-guess examples (e.g., "The Apple Watch was created by _" with the answer *Apple*). We test the 5-shot accuracy of 9 popular LMs of various sizes: GPT-3 [175B, 13B, 6.7B, 2.7B parameter models; 2], GPT-2 [1.5B, 782M, 345M, 117M models; 1], and DistilGPT-2 [16], a distilled, 82M parameter version of GPT-2 117M.[2] The supplementary material contains the code to reproduce all results and plots in our paper.

**Prompts**    To form our set of candidate prompts $\mathcal{A}_1, \ldots, \mathcal{A}_A$, we rely on LAMA as well as the LM Prompt And Query Archive [LPAQA; 4]. For each relation, we use the manually-written prompt from LAMA, plus LPAQA prompts formed by (1) paraphrasing the manual prompt with back-translation (2) mining from Wikipedia, and (3) paraphrasing the top mined prompt. For each relation, we use up to 16 prompts with a mean of 12 prompts (see Appendix §E.1 for details on the prompts we use).

**Computing CV and MDL**    As the loss function $\mathcal{L}$, we use the negative log-likelihood (NLL) of the label given the input over all held-out examples $\sum_{(x,y)} -\log p(y|x)$. We use NLL as in prior work on MDL [32, 61, 10], to retain MDL's property as a measure of label compression. For CV, NLL avoids ties between prompts that arise from using accuracy in the context of such limited data (e.g., 5 examples). We use $K = N$ folds (where $N$ is the number of training examples) for both MDL and CV (here, LOOCV). Here, $N$-fold CV requires $N$ forward passes to evaluate the loss on each of the $N$ examples when conditioning on the $N - 1$ other examples. $N$-fold MDL can be computed using a

---

[2]We use OpenAI's API for GPT-3 (`https://beta.openai.com/`) and HuggingFace Transformers [59] via PyTorch [60] for GPT-2/DistilGPT-2. OpenAI does not disclose API model sizes, so following [6, 7], we assume that the 4 API models are the 4 largest from [2]. We will update our paper should OpenAI release model sizes.

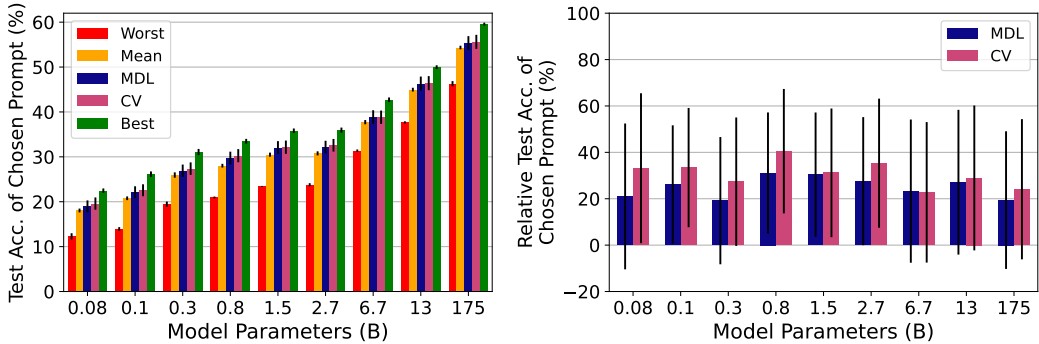

Figure 1: **Left**: LAMA-UHN accuracy of CV/MDL-chosen prompts vs. accuracy of the worst, average (randomly-selected), and best prompt (prior work). **Right**: The average accuracy gain from using CV/MDL-chosen prompts instead of randomly-chosen ones, relative to the gain from the best prompt. We plot mean/std. err. across 5 runs with different training sets. CV/MDL-chosen prompts obtain only small improvements over randomly-chosen ones and do far worse than the best prompts.

single LM forward pass to compute the loss on each example conditioned on the previous examples. This feature makes MDL more computationally efficient than CV.

**Marginalizing out example order**    The order of training examples impacts the generalization of LMs [7], so we treat order as a random factor $R$ that we marginalize over to evaluate the generalization of a prompt $\mathcal{A}$. We compute the exact $\mathbb{E}_{R,F}[\text{CV}(\mathcal{A}, R, F)]$ and $\mathbb{E}_{R,F}[\text{MDL}(\mathcal{A}, R, F)]$ by averaging over all $N!$ training example orders. We use $N = 5$ examples to limit $N!$. We estimate the average test accuracy on $N! = 120$ examples in LAMA, excluding the training examples, by evaluating on one test example per permutation of training examples. We compute CV, MDL, and test accuracy with $N! = 120$ forward passes in total by appending a test example to each permutation of training examples, and we compute all selection criteria using the same set of $N! = 120$ forward passes to maximize comparability across different methods. We show the test accuracy from CV/MDL-chosen prompts, averaged over all relations. For comparison, we show the test accuracy of always choosing (1) the best prompt, chosen using held-out accuracy as in prior work, (2) the worst prompt, as a lower bound, and (3) random prompts (we show the mean accuracy over all prompts).

### 3.2    How well does prompt selection do in true few-shot learning?

Fig. 1 (left) shows the results; prompt selection obtains marginal improvements over random selection across model sizes ranging over 3 orders of magnitude. Prompts chosen by CV and MDL alike underperform the best prompt (chosen using held-out performance) by 5-7% absolute on average. In fact, prompts chosen based on held-out performance often outperform larger models whose prompts are chosen in a true few-shot manner. CV and MDL do tend to choose better-than-average prompts, but only close the gap between the average and best prompts by 20-40%, as shown in Fig. 1 (right). We find similar results for several other tasks (§3.7) and selection criteria (Appendix §B).

Fig. 2 (left) shows that CV/MDL struggle to choose the prompt with the highest test accuracy. Poor top-prompt selection is especially prevalent for larger models like GPT-3 175B that have spurred interest in prompt design (only 21% accuracy for CV vs. 9% for random chance). In Appendix §A, we show that CV/MDL-chosen prompts differ from the best prompts not only in accuracy but also in how effectively they transfer to models of different sizes. Altogether, our results show that optimal prompt selection is difficult in the true few-shot setting, and that prior work overestimated the ability of LMs by using held-out examples for prompt selection.

### 3.3    How reliably does prompt selection improve over the average prompt?

If the average improvement from prompt selection is small, can we at least obtain an improvement with high probability for a given task and train set? Fig. 1 (left) shows that the worst prompts do far worse than average, so it would be useful if prompt selection helped to avoid the worst prompts. We

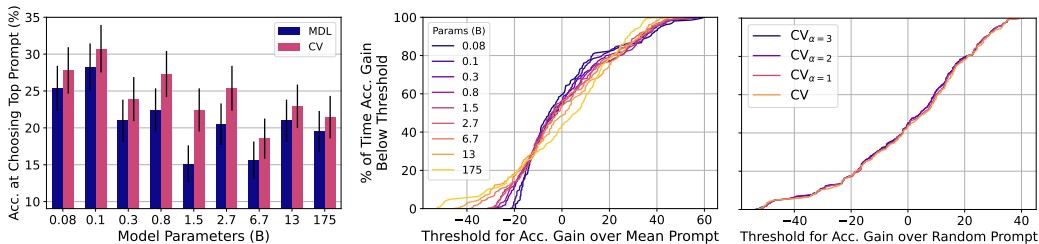

Figure 2: **Left**: CV/MDL have low accuracy at choosing the best prompt (mean/std. err. across 5 runs with different train sets). **Middle**: The chance of various accuracy gains over the average prompt, when using prompts chosen by CV, and (**Right**) conservative estimates of CV that also minimize variance in CV; CV often chooses worse-than-average prompts, even with conservative selection.

examine the probability with which prompt selection obtains various accuracy gains over the average prompt and show results in Fig. 2 (middle) for CV (and similar results in Appendix §C for MDL).

CV/MDL-chosen prompts show high variance in test accuracy relative to the average prompt. For most model sizes (.1B-6.7B), the chance of improving over the average, randomly-chosen prompt is only ~56% for CV and ~55% for MDL. The performance of prompt selection forms a long-tailed distribution; there is a ~27% chance that prompt selection causes an accuracy drop of ~13% for all model sizes and CV/MDL alike. Furthermore, the tails grow heavier as model size increases. For the largest model (GPT-3 175B), CV/MDL-chosen prompts sometimes do far worse than average, e.g., 40% worse, 5% of the time. Our results suggest a troubling trend: as models grow bigger and generalize better, our ability to reliably choose good prompts degrades. One possible explanation is that larger models have the capacity to draw more complex decision boundaries, requiring more examples to estimate the true expected loss on unseen examples; we may need to scale validation sets along with model size. In §3.7, we show similar results for several other tasks. Overall, the limited average-case gains from prompt selection cannot be expected with any reasonable confidence in the true few-shot setting, a problem that will only become worse with larger models.

### 3.4 Can we increase the likelihood of improved performance from prompt selection?

As we have shown, CV and MDL do not reliably choose better-than-average prompts. Here, we explore the extent to which we can reduce the variance in generalization by explicitly preferring prompts with low variance (§2.3). For the largest model (GPT-3 175B), we choose prompts based on a conservative estimate of generalization loss, $CV_\alpha$ (§2.3). We show the test accuracy for the prompt chosen with various levels of confidence $\alpha \in \{1, 2, 3\}$ and with CV ($\alpha = 0$).

As shown in Fig. 2 (right), all $\alpha$ lead to a similar distribution of performance gain as CV. For example, CV outperforms the average prompt 50% of the time vs. 51% for $\alpha = 2$. These results suggest that it is non-trivial to choose prompts that reliably perform better than random selection, even when explicitly minimizing variance in generalization.

### 3.5 Does prompt selection improve with more labeled examples?

The poor performance of prompt selection methods may be due to using such a small number of labeled examples. As the number of labeled examples increases, we expect prompt selection methods to improve. Thus, true few-shot prompt selection may be possible with a few dozen examples (though it is not always possible to use more examples, due to limits on input length for LMs like GPT). We therefore examine the test accuracy of CV/MDL-chosen prompts as we use an increasing number of labeled examples $N \in \{5, 10, 15, 20, 30, 40\}$. For $N \geq 10$, it is not feasible to marginalize over all possible training example permutations, so we randomly sample 120 permutations (to match $N = 5$) such that each example occurs the same number of times in each position (i.e., to use each example as the held-out CV fold the same number of times). We run the experiment for $\leq$6.7B parameter models, since it is prohibitively costly to run with larger models via the OpenAI API.

As shown in Fig. 3, there is no consistent trend in the performance of prompt selection, both in terms of task performance (left) and in terms of accuracy at choosing the highest accuracy prompt (right).

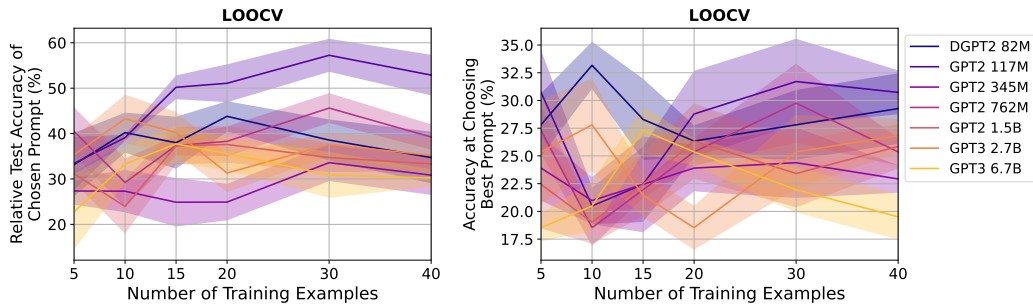

Figure 3: Increasing the number of examples up to 40 does not clearly improve CV in terms of (**Left**) accuracy gain over the average prompt (scaled to 0), relative to the best one (scaled to 100) or (**Right**) accuracy at choosing the best prompt. Mean/std. err. on LAMA over 5 runs (varying train sets).

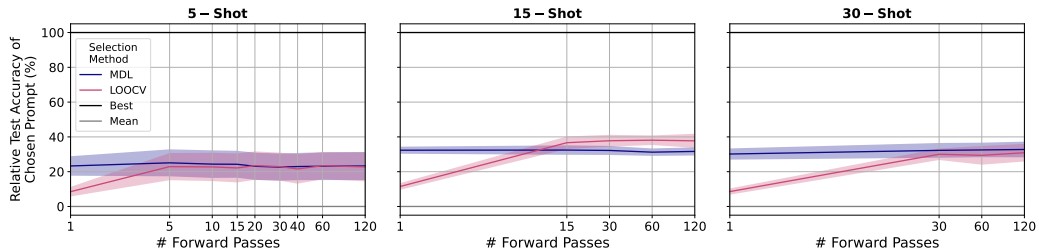

Figure 4: For $N \in \{5, 10, 30\}$ -shot learning, increasing the compute used to estimate CV/MDL does not notably improve the accuracy of chosen prompts beyond a certain point (1 forward pass for MDL, $N$ forward passes for CV). Mean/std. err. across 5 runs for GPT-3 6.7B.

Even in higher-data regimes (40 examples), CV/MDL struggle to choose effective prompts and do not consistently, across model sizes, perform better than choosing examples based on 5 examples. Our findings are surprising, because the true-few shot setting is where prompt design has been thought most promising, due to the scarcity of training data [15]. However, the true few-shot setting is also one in which prompt selection is hardest, greatly undermining the potential value of prompts.

### 3.6  Does prompt selection improve with more computation?

In the preceding sections, we computed $\mathbb{E}_{R,F}[\mathrm{CV}(\mathcal{A}, R, F)]$ using a fixed number of samples for $R$. Can we improve prompt selection by using more samples, at the cost of increased computation? To answer this question, we vary the number of samples of $R$ (and thus LM forward passes) used to compute the above expectation and choose prompts as described in §2.3. To estimate CV with a single forward pass, we sample a single fold $k$ (here, a single example) and evaluate accuracy on fold $k$ when conditioning the LM on all others folds. Fig. 4 shows the results for $N \in \{5, 15, 30\}$ training examples using the largest model from §3.5 (GPT-3 6.7B).

Computation is not the bottleneck in prompt selection, as test accuracy roughly plateaus after one forward pass for MDL and $N$ forward passes for CV. This observation holds across $N$, as well as all models with <6.7B parameters (omitted for space). Our results suggest that true few-shot prompt selection is fundamentally limited by the number of examples available.

### 3.7  Is Prompt Selection Challenging on Other Tasks?

We now examine the extent to which our results on LAMA tasks hold on other kinds of NLP tasks. We examine three classification tasks for which prior work has designed various prompts: Recognizing Textual Entailment (RTE), CommitmentBank (CB), and Word-in-Context (WiC). RTE and CB involve detecting whether one sentence entails or contradicts another, and WiC involves determining if a polysemous word is used with the same sense in two sentences (e.g., "Room and board" and "He nailed boards across the windows."); See Appendix§E.2 for further task details. We evaluate the accuracy of GPT models when using prompts chosen by CV, MDL, and test accuracy, as we did

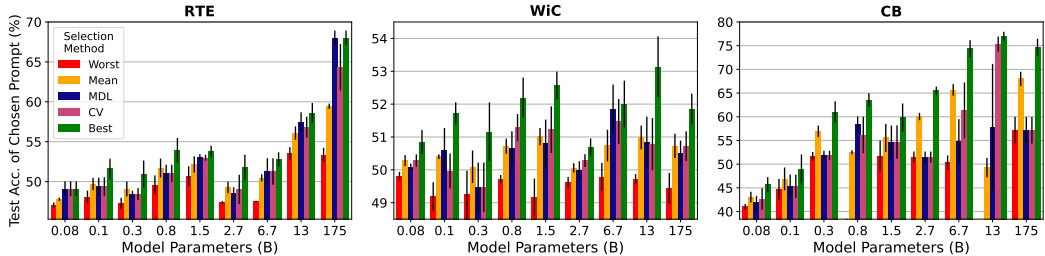

Figure 5: Accuracy of CV/MDL-chosen prompts vs. accuracy of the worst, average (randomly-selected), and best prompt (prior work), on three classification tasks (mean/std. err. over 5 runs). CV/MDL-chosen prompts generally perform several points worse than the best prompt and do not consistently improve over the average prompt across tasks and model sizes.

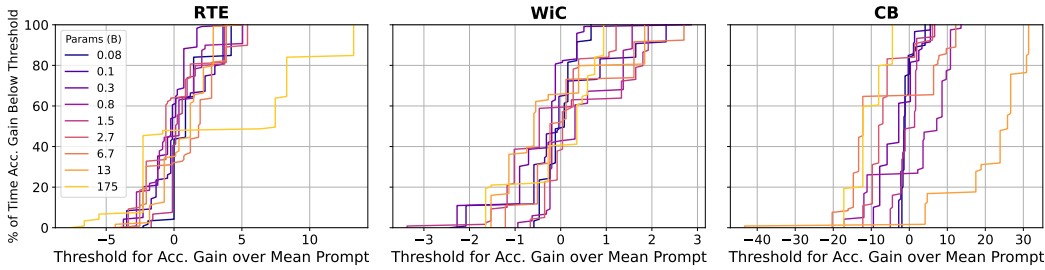

Figure 6: The chance of various accuracy gains over the average prompt from CV on RTE, WiC, and CB. CV often chooses prompts that are below average (RTE, WiC) or far below average (CB).

for LAMA. For each task, we evaluate held-out accuracy using the full validation set when using 5 training examples randomly sampled from the task train set, while ensuring that we include at least one example per class. We evaluate the mean/std. error over 5 train sets. As our set of prompts, we use the manually-written prompts from [2] and [9] – 3 prompts for RTE/CB and 4 prompts for WiC. Schick and Schütze [9] designed prompts for bidirectional LMs, so when necessary, we modify their prompts to be suitable for left-to-right LMs (see Appendix §E.2 for prompts). Fig. 5 shows the accuracy of the chosen prompts on each task.

We observe as similar trend as before, that across tasks and model sizes, the CV/MDL-chosen prompt almost always obtains lower average accuracy than choosing based on test accuracy. The trend holds even when choosing between fewer prompts (here, 3-4). CV/MDL-chosen prompts vary greatly in test accuracy across tasks and model sizes, often choosing worse-than-average prompts (e.g., on CB).

We examine the variance in chosen prompt accuracy in more detail, by showing the chance that selection obtains various accuracy gains over the average prompt. Here, we choose prompts with CV using $N$ forward passes (one evaluation per fold), as it represents a good tradeoff between compute and accuracy that is likely to be used in practice. As shown in Fig. 6, accuracy gains are again highly dispersed, often negative, and not consistently achieved. For CB, there is a 20% change of a 15% accuracy drop for GPT-3 175B. Model sizes vary greatly in how often the CV-chosen prompt leads to improvement, e.g., from 38-82% for WiC and 1-83% for CB. Overall, our earlier findings carry over to other kinds of tasks, indicating that prompt selection is challenging in general.

## 4 True Few-Shot Hyperparameter Selection

Having shown that true few-shot prompt selection is difficult, we now study the effectiveness of model selection in the context of hyperparameter selection more generally. We examine the ADAPET model [12], as it is open-source[3] and currently the top-performing few-shot model according to SuperGLUE [62], a standard NLP benchmark. ADAPET finetunes the pretrained ALBERT_xxlarge-v2 LM [63] to (1) classify each label as correct/incorrect given the input and (2) to predict randomly

---

[3] https://github.com/rrmenon10/ADAPET

| | BoolQ
Acc | CB
Acc/F1 | COPA
Acc | RTE
Acc | WiC
Acc | WSC
Acc | MultiRC
EM/F1 | ReCoRD
EM/F1 | Avg |
|---|---|---|---|---|---|---|---|---|---|
| **Worst** | $75.0_{4.8}$ | $79.5_{2.3}/67.3_{7.8}$ | $76.8_{2.2}$ | $63.2_{4.0}$ | $49.0_{1.3}$ | $77.2_{1.8}$ | $38.5_{7.4}/80.0_{2.9}$ | $76.2_{1.8}/86.5_{1.2}$ | $69.4_{1.5}$ |
| **Mean** | $79.0_{1.5}$ | $85.9_{2.3}/74.5_{11.0}$ | $81.1_{2.9}$ | $70.8_{2.5}$ | $51.5_{1.8}$ | $82.5_{2.7}$ | $44.2_{6.6}/82.3_{2.7}$ | $78.3_{1.3}/87.8_{0.8}$ | $73.9_{1.2}$ |
| **MDL** | $76.5_{5.8}$ | $85.7_{5.6}/74.8_{13.4}$ | $82.0_{2.9}$ | $70.4_{8.5}$ | $52.2_{3.0}$ | $82.0_{3.1}$ | $39.7_{8.1}/80.6_{3.2}$ | $78.9_{0.7}/88.2_{0.4}$ | $73.4_{2.8}$ |
| **CV** | $78.9_{2.4}$ | $83.9_{5.3}/69.2_{10.3}$ | $80.5_{3.3}$ | $68.7_{7.0}$ | $51.1_{1.6}$ | $83.1_{2.6}$ | $41.9_{7.2}/81.4_{3.1}$ | $78.7_{1.6}/88.1_{1.0}$ | $73.0_{2.1}$ |
| **Best** | $80.9_{1.0}$ | $89.8_{3.1}/79.8_{13.4}$ | $84.8_{4.5}$ | $76.7_{1.8}$ | $54.1_{2.3}$ | $86.6_{1.8}$ | $46.8_{6.9}/83.4_{2.9}$ | $80.4_{1.1}/89.2_{0.7}$ | $77.2_{0.9}$ |
| **ADAPET** [12] | 80.3 | 89.3 / 86.8 | 89.0 | 76.5 | 54.4 | 81.7 | 39.2 / 80.1 | 85.4 / 92.1 | 77.3 |
| **iPET** [9] | 80.6 | 92.9 / 92.4 | 95.0 | 74.0 | 52.2 | 80.1 | 33.0 / 74.0 | 86.0 / 86.5 | 76.8 |
| **PET** [9] | 79.4 | 85.1 / 59.4 | 95.0 | 69.8 | 52.4 | 80.1 | 37.9 / 77.3 | 86.0 / 86.5 | 74.1 |
| **GPT-3** [2] | 77.5 | 82.1 / 57.2 | 92.0 | 72.9 | 55.3 | 75.0 | 32.5 / 74.8 | 89.0 / 90.1 | 73.2 |

Table 1: ADAPET results on SuperGLUE validation when choosing early stopping checkpoint and masked LM rate using CV/MDL vs. the worst/mean/best hyperparameters chosen with validation (mean$_{\text{std. dev.}}$ over four 32-shot train sets). On all tasks, CV/MDL-chosen hyperparameters perform similar to or worse than average, and several points below the best hyperparameters.

masked out input tokens given the label and unmasked input tokens, similar to Masked LM [64]. ADAPET was developed in the context of tuned few-shot learning, as ADAPET's hyperparameters were chosen using validation examples. We examine how ADAPET does in the true few-shot setting.

We choose two hyperparameters in a true few-shot manner: the early stopping checkpoint and fraction of words masked for the masked LM objective. ADAPET performs $T = 1000$ gradient updates on batches of 16 examples and chooses the checkpoint at $T \in \{250, 500, 750, 1000\}$ with the highest validation accuracy. ADAPET also chooses the best masking fraction $M \in \{0.075, 0.10, 0.105, 0.15\}$. Following ADAPET, we evaluate on SuperGLUE, a suite of 8 NLP tasks. SuperGLUE consists of four question-answering tasks (BoolQ, COPA, MultiRC, ReCoRD), a coreference resolution task (WSC), two entailment detection tasks (RTE, CV), and a commonsense reasoning task (WiC); see Appendix §E.2 for task details. We use CV/MDL to choose $T$ and $M$ (out of 16 total combinations) and then train a model on the full dataset with the chosen $T$ and $M$. We use FewGLUE [9], the 32-example subset of SuperGLUE used in prior work on few-shot learning. We also use 3 other 32-example subsets that we randomly sample from SuperGLUE, to estimate variance in performance across training sets. ADAPET uses a prompt during fine-tuning, choosing the prompt based on validation examples. To avoid using validation-tuned prompts, we use the first prompt for every task as the authors do for ablation studies. Since training ADAPET is expensive, we evaluate CV/MDL with $K = 8$ folds.[4] We show results in Table 1.

**Results** Across all SuperGLUE tasks, CV/MDL hyperparameter selection performs similar to or worse than average (randomly-chosen) hyperparameters and several points worse than the best hyperparameters. In the true few-shot setting, the average SuperGLUE performance of ADAPET drops below that of earlier methods (PET and iPET), highlighting how the use of validation examples can give the false appearance of progress in few-shot learning. On MultiRC, CV/MDL choose hyperparameters that give similar performance to the worst hyperparameters, another indication that model selection methods do not consistently prevent worst-case behavior in the true few-shot setting. Preliminary analysis in Appendix §D suggests that choosing better-than-average hyperparameters requires several thousand examples. Overall, our results indicate that it is not just prompt selection but model selection in general that is challenging in very low-data regimes.

## 5 Conclusion and Future Work

Our work shows that it is challenging to make even the most basic decisions about few-shot learning algorithms using only a few labeled examples. Instead, it may be more promising to make additional assumptions. The meta-learning setting assumes access to data from many other tasks in order to perform learning and model selection [20, 21, 65, 66]. Transfer learning and multitask learning assume access to data that is directly related to the task with limited data [67–70]. Data augmentation techniques assume there is a viable way to create more data from limited data [71–74]. Other approaches assume unlabeled data and develop unsupervised model selection techniques [75–77]. When labeled data is cheap, the simplest approach is to assume more examples for validation—

---

[4]See Appendix §E.4 for details on how we evaluate MDL on different SuperGLUE tasks.

in which case we might be better off training on the additional examples. Unless we make such assumptions explicit, we cannot make meaningful comparisons between few-shot learning algorithms. We find the above avenues to be more promising future directions than true few-shot learning given the challenge of model selection.

Inspired by prior work [78–81], we offer recommendations for future work in true few-shot learning:

- Report all hyperparameters (prompts) considered and the hyperparameter selection criteria.

- Include validation examples in the number of examples used by a few-shot learning algorithm. Validation examples include all examples used to decide on any aspect of learning: hyperparameters, prompts, training objectives, decoding strategies, model architecture, etc.

- Once you have decided on the learning algorithm, submit your model for test evaluation directly, without first evaluating on validation. Report the total number of test evaluations conducted (ideally, just one). Use the validation set only after test evaluation for any ablations you report, to avoid making decisions about your algorithm with the validation set.

- Do not rely on hyperparameters from prior work that were tuned using validation examples for the same benchmark (e.g., SuperGLUE), to avoid indirectly benefiting from validation examples. Instead, re-tune such hyperparameters using only the given few-shot examples.

The above protocols are strict but mimic how a true few-shot learning algorithm would be used in a real, low-data setting. To ensure researchers comply with such strict protocols, future benchmarks may need to keep large test sets private while releasing only a few labeled examples.

Given our negative results on true few-shot learning, a major question remains: is it possible to select models in a true zero-shot setting? Prior work uses LMs for zero-shot learning by choosing an arbitrary prompt [17, 82] which requires no data but is suboptimal [4]. Other efforts try multiple prompts and choose between them via trial and error alongside manual evaluation [1], effectively leveraging human supervision. CLIP [13] achieves high zero-shot accuracy on ImageNet after extensively tuning prompts and label names using ImageNet's training set (1.28M examples), as we noticed from the open-source code.[5] The authors report a 5% accuracy gain from tuning prompts alone, but the training examples used for tuning are not available in true zero-shot learning. Without any labeled data, the problem of model selection is even more challenging than in the true few-shot case. Overall, our work provides guidance for future work in few-shot learning by clarifying the assumptions made by the true few-shot setting and empirically demonstrates that model selection is a major roadblock to true few-shot learning.

## 6   Limitations and Broader Impact

We facilitate fair comparisons between few-shot methods in future work by disambiguating between three few-shot settings: multi-distribution, tuned, and true few-shot learning. We highlight that one setting, tuned few-shot learning, gives up the practical advantage of few-shot learning by using many labeled examples. Furthermore, we show that several tuned few-shot learning algorithms work significantly worse in the true few-shot setting, without tuning on many examples. While we explored CV, MDL, and several other criteria (Appendix §B), it is possible that effective true few-shot model selection is possible using other criteria (§2.4) or heuristics not explored here. In this event, our work will have discouraged work on a few-shot learning setting with applications to low-data settings, e.g., that involve low-resource languages or expert annotation. Overall, however, we believe our work will redirect future work to few-shot settings with more practical applications.

We show that it is hard to detect when a small input change hurts an LM's generalization, even when the change appears reasonable to human readers. We argue that practitioners will benefit from knowing such limitations, but they may also be discouraged from deploying LMs in many useful contexts, such as question-answering, hate speech detection, automatic translation, and commercial dialogue systems. Our findings may also encourage adversaries to target LM-based applications and highlight which models are most susceptible to attack (e.g., larger models). By shedding light on the shortcomings of (few-shot) LMs, we hope to spur future work to address these shortcomings.

---

[5]`https://github.com/openai/CLIP/blob/main/notebooks/Prompt_Engineering_for_`
`ImageNet.ipynb`

## Acknowledgments

We are grateful to OpenAI for providing access and credits to GPT-3 via the API Academic Access Program, and we thank Miles Brundage, David Schnurr, Felipe Such, Ryan Lowe, and Ilya Sutskever for help with the API. We thank GPT-3 authors Benjamin Mann and Gretchen Krueger for helpful feedback on our paper. We thank Rakesh Menon for assistance with the ADAPET codebase, Shenglong Wang for cluster support, Zhengbao Jiang for LPAQA prompts, and Tal Linzen, Patrick Lewis, Eric Wallace, Adam Fisch, Stephen Roller, Aravind Rajeswaran, Gretchen Krueger, Amanda Ngo, Udit Arora, Sébastian Jean, Jason Phang, and the NYU NLP group for feedback on our draft. KC is partly supported by Samsung Advanced Institute of Technology (Next Generation Deep Learning: from pattern recognition to AI) and Samsung Research (Improving Deep Learning using Latent Structure). KC also thanks Naver, eBay, NVIDIA, and NSF Award 1922658 for support. EP is grateful to NSF and Open Philanthropy for fellowship support.

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
