

Figure 7: A model's accuracy with the prompt chosen for another model using MDL, CV, or test accuracy. We show LAMA accuracy relative to the average prompt (scaled to 0) and best prompt (scaled to 100) for a model size. CV/MDL show different patterns in prompt transfer than test acc.

# A To What Extent Are Chosen Prompts Specific to the Model?

We investigate the extent to which CV/MDL-chosen prompts differ from the best, test-chosen prompts in other ways, aside from accuracy. To this end, we examine how well a model does when using a prompt chosen for another model, which we refer to as "prompt transfer." Prompt transfer indicates how tailored the chosen prompt is to a given model. For each model, we examine the average gain of the chosen prompt over the average prompt, relative to the maximum possible gain, i.e., scaling the test accuracy for each model so that the average prompt scores 0% and the top prompt scores 100%.

As shown in Fig. 7, prompts chosen based on test accuracy generalize reasonably well across models of similar sizes, a pattern that degrades as we examine CV and especially MDL. For CV, prompts chosen using one model size do transfer better to similar model sizes, but CV-chosen prompts do not transfer as effectively as test-chosen ones. For MDL, the chosen prompts are not particularly tailored to the given model, performing similarly across many model sizes. Overall, even the pattern of prompt transfer differs between test accuracy and CV/MDL.

# B True Few-Shot Prompt Selection with Other Generalization Criteria

Here, we evaluate the performance of prompts chosen using other generalization criteria, to examine the extent to which poor prompt selection is specific to CV and MDL. We evaluate on LAMA and follow the same experimental setup used to evaluate CV/MDL, as described in §3.1. As before, we examine the average test accuracy of the prompt chosen by a particular criterion, as well as the percentage of the time that a given criterion chose the prompt with the highest test accuracy. We now describe the other criteria we test.

## B.1 Bayesian Cross-Validation

Bayesian CV is a variant of CV that evaluates a learning algorithm $\mathcal{A}$ based on its expected loss on a held-out fold after marginalizing over the model according the posterior distribution [for an overview, see 83]. In our setup, each model corresponds to a unique set of random factors $R$ trained by $\mathcal{A}$. Given some inputs $X = x_{1:N}$ and labels $Y = y_{1:N}$, we assume a uniform prior $p(R)$ over $R$ and assume that $R$ and $X$ are independent ($p(R|X) = p(R)$). We then derive the posterior probability as:

$$p(R|X,Y) = \frac{p(Y|R,X)p(R|X)}{p(Y|X)} = \frac{p(Y|R,X)}{p(Y|X)} = \frac{p(Y|R,X)}{\sum_{R'} p(Y|R',X)}$$

where for any $R'$:

$$p(Y|R',X) = \prod_{i=1}^{N} p(y_i|y_{1:i-1}, X, R') = \prod_{i=1}^{N} p(y_i|y_{1:i-1}, x_{1:i}, R').$$

The second equality holds because $p$ is a left-to-right LM that predicts $y_i$ only based on the input $x_i$ and earlier examples $(x_{1:i-1}, y_{1:i-1})$. We marginalize out the model over the posterior distribution:

$$\text{CV}_{\text{Bayes}}(\mathcal{A}, R, F) = \mathbb{E}_{k \sim \text{Unif}(1,K)}\left[\mathcal{L}\Big(\mathbb{E}_{R \sim p(R|F(D_{\text{train}})_{\neg k})}[\mathcal{A}(F(D_{\text{train}})_{\neg k}, R)]; F(D_{\text{train}})_k\Big)\right]$$

We then choose the algorithm (prompt) that minimizes $\mathbb{E}_{R,F}[\text{CV}_{\text{Bayes}}(\mathcal{A}, R, F)]$, where $R$ is the order of training examples.

## B.2 Interpolating between CV and MDL

Our experiments in the main paper suggest that CV/MDL behave differently in terms of prompt selection. In this section, we describe a way to interpolate between CV and MDL, in order to devise a new criterion that may inherit advantageous properties from both CV and MDL. Similar to MDL, we measure the expected loss on a held-out fold $F(D_{\text{train}})_k$ when training on the previous $F(D_{\text{train}})_{1:k-1}$ folds, doing so across all $k = 1, \ldots, K$ folds. However, we now weight the loss on $F(D_{\text{train}})_k$ by a factor that depends on the number of training examples, $p(k; \beta) \propto \exp(-\beta|F(D_{\text{train}})_{1:k-1}|)$, where $\beta$ is an inverse temperature hyperparameter. MDL is equivalent to using a uniform weight over all train sizes ($\beta = 0$), and CV is equivalent to using a non-zero weight for only the largest train size ($\beta = \infty$). Formally, we define the interpolated criteria, $\text{MDL}_\beta(\mathcal{A}, R, F)$, as follows:

$$\text{MDL}_\beta(\mathcal{A}, R, F) = \mathbb{E}_{k \sim p(k; \beta)}\left[\mathcal{L}\Big(\mathcal{A}(F(D_{\text{train}})_{1:k-1}, R); F(D_{\text{train}})_k\Big)\right].$$

We set the hyperparameter $\beta$ to the default value of $\beta = 1$ to avoid having to choose $\beta$ based on a limited number of examples available in true few-shot learning. We choose the algorithm that minimizes $\mathbb{E}_{R,F}[\text{MDL}_\beta(\mathcal{A}, R, F)]$.

## B.3 Joint Log-Probability

Up to this point, we have used generalization criteria that use the NLL of the label given the input, $-\log p(y|x)$, as the loss function $\mathcal{L}$. However, other loss functions may correlate better with generalization. In particular, we hypothesize that a good prompt leads the LM to give the entire input $(x, y)$ high probability, i.e., a low, joint log-probability $-\log p(x, y)$. We thus use $-\log p(x, y)$ as the loss function to measure CV and MDL, which we refer to as $\text{CV}_{x,y}$ and $\text{MDL}_{x,y}$, respectively. Since $-\log p(x, y) = [-\log p(y|x)] + [-\log p(x)]$, joint log-probability is equivalent to the label NLL $-\log p(y|x)$ used before, with an additional term $-\log p(x)$ that measures the input NLL. We measure $-\log p(x, y)$ by evaluating the total NLL of all tokens in the prompt-formatted $(x, y)$ pair (including prompt tokens). We choose the algorithm that minimizes $\mathbb{E}_{R,F}[\text{CV}_{x,y}(\mathcal{A}, R, F)]$ or $\mathbb{E}_{R,F}[\text{MDL}_{x,y}(\mathcal{A}, R, F)]$.

## B.4 Results

As shown in Fig. 8 (top), all criteria choose prompts with a similar average accuracy, close to the average accuracy of randomly-chosen prompts. Likewise, all criteria are similarly inaccurate at choosing the highest accuracy prompt, as shown in Fig 8 (bottom). These results show that true few-shot prompt selection is challenging not only for CV and MDL but also many other criteria.

## C Additional Results with MDL

In the main paper, we showed several results for CV alone for brevity, so in this section, we show the corresponding plots for MDL as well. The overall trends are the same for both CV and MDL.

In §3.3, we found that the gains from choosing prompts using CV are high variance, a variance that increases with model size (Fig. 2). Here, we show the same results but for MDL in Fig. 9 (left). Similar to CV, MDL-chosen prompts have high variance in test accuracy relative to the average prompt, especially for larger models. This finding suggests that the high variance is due not to CV in particular, but to the inherent difficulty of true few-shot model selection.

In §3.5, we examined if increasing the number of examples improves prompt selection for CV. Fig. 9 (middle/right) shows the results for MDL, which are similar to those for CV. When increasing the

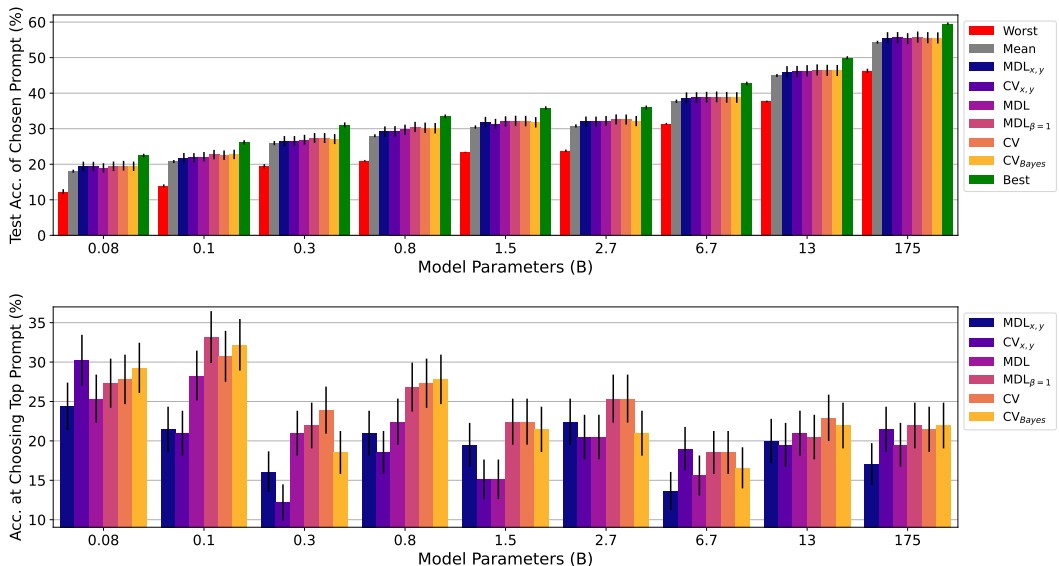

Figure 8: **Top**: LAMA-UHN accuracy of prompts chosen using different generalization criteria vs. accuracy of the worst, average (randomly-selected), and best prompt (prior work). **Bottom**: The average accuracy gain from using criteria-chosen prompts instead of randomly-chosen ones, relative to the gain from the best prompt. We plot mean/std. err. across 5 runs with different training sets. Across all model sizes, criteria-chosen prompts obtain only small improvements over randomly-chosen ones and perform far worse than the best prompts.

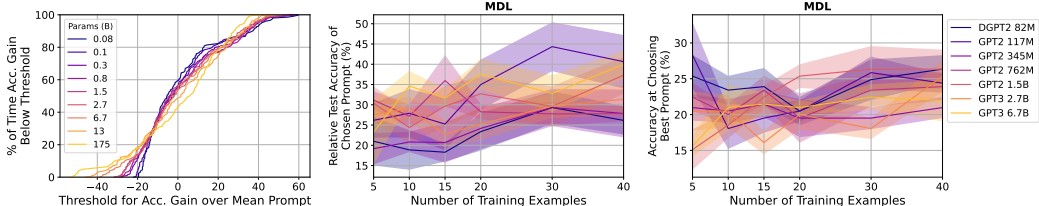

Figure 9: **Left**: Chance of various accuracy gains for MDL-chosen prompts over average (randomly-chosen) prompts on LAMA-UHN. As with CV, there is a wide variance in accuracy gains, especially for larger models, and a significant chance of choosing a worse-than-average prompt. **Middle**: Increasing the number of examples up to 40 does not clearly improve MDL in terms of acc. gain over the average prompt (scaled to 0), relative to the best one (scaled to 100) or (**Right**) acc. at choosing the best prompt (mean/std. err. on LAMA over 5 runs with different train sets).

examples used, we do not observe a consistent increase in the gain achieved by MDL over random selection, relative to the best prompt (Fig. 9 middle). Similarly, we do not observe a consistent increase in the accuracy of MDL at choosing the best prompt (Fig. 9 right). For some model sizes, there may potentially be some improvement with more examples, but the standard error is high, and the overall accuracies achieved by MDL are still lower than those from CV shown earlier in Fig. 3. Overall, model selection is challenging for both CV and MDL, even as we approach the maximum number of examples that can fit in the context of GPT models.

# D    How Many Examples Do You Need for Effective Model Selection?

Here, we conduct a preliminary analysis on the minimum number of examples is necessary to choose a better-than-average model. We examine this question in the context of ADAPET, which can handle an arbitrary number of examples (GPT-based models can only handle a number of examples that fit within the LM input—2048 tokens or ~1500 words). We use the same setup and hyperparameter range as in §4 but vary the number of training examples.

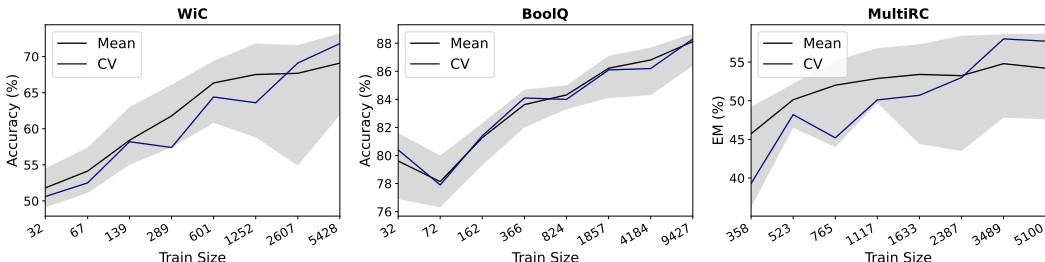

Figure 10: ADAPET accuracy using CV-chosen hyperparameters as the number of examples increases. The shaded region shows the range of accuracies obtained using the same training set but different hyperparameter settings (16 in total).

Fig. 10 shows accuracy on WiC and BoolQ of CV-chosen hyperparameters, compared to the worst, average, and best hyperparameters. For WiC and MultiRC, CV requires >2-3k examples to choose better-than-average hyperparameters. For BoolQ, CV performs similar to the average hyperparameters even when using up to 9k examples. This result may be due to the fact that we retrain the model using the CV-chosen hyperparameters, but finetuning pretrained LMs often has high variance in performance [69, 84]. Thus, when more data is available, CV may be outperformed by using a single train-validation split and choosing the model that does well on the validation split, without retraining on the combined train+validation set. We leave further exploration of model selection in higher data regimes as an important direction for future work.

# E    Task and Experimental Details

## E.1    LAMA

**Prompts Used**    For the full list LPAQA prompts, please see `https://github.com/jzbjyb/LPAQA/tree/master/prompt`. There are up to 90 LPAQA prompts per relation, so we use a subset of prompts to evaluate the impact of a small amount of validation-based prompt tuning. We filter out prompts that do not end with the target answer blanked out ("Geoffrey Hinton was _ profession."), which cannot be easily used with left-to-right LMs like GPT. For mined prompts (group 2), we choose the 5 prompts that occur most frequently in Wikipedia, similar to [4]. We include all prompts if fewer than 5 are available. For paraphrased prompts (groups 1 and 3), we choose up to 5 prompts with the highest round-trip back-translation probability, similar to [4]. Finally, we de-duplicate prompts, as some prompts occur in multiple groups.

**Few-shot Learning**    LAMA is typically used to evaluate LM knowledge in a zero-shot rather than few-shot way. Thus, it is important to test that LMs benefit from few-shot examples on LAMA. We find that GPT3 175B achieves 22% accuracy on LAMA in the zero-shot setting vs. 54% in the 5-shot setting, suggesting that LMs do benefit from few-shot examples on LAMA. Our result suggests that LAMA is an appropriate benchmark for evaluating few-shot learning in LMs.

## E.2    SuperGLUE

**Datasets**    Here, we go into more detail about various tasks in SuperGLUE [62]. BoolQ [Boolean Questions; 85] involves answering a yes/no question about a paragraph. COPA [Choice of Plausible Alternatives; 86] involves determining the cause (or effect) of a given premise from two possible choices. RTE (Recognizing Textual Entailment) is a 2-sentence classification task to determine if a given premise entails a given hypothesis (2-way classification between entailed and not entailed classes) [87–90]. Similarly, CB [CommitmentBank; 86] is an entailment detection task but with 3 classes (entailed, contradicted, and neither). WiC [Word-in-Context, 91] involves determining if a polysemous word is used with the same sense in two sentences. WSC [Winograd Schema Challenge, 92] is a coreference resolution task to determine the correct referent of a pronoun in a sentence from among the provided choices. MultiRC [Multi-Sentence Reading Comprehension, 93] is a question-answering task where each example consists of a context paragraph, a question about that paragraph, and a list of possible answers, multiple of which can be correct. ReCoRD [Reading

| Task | Prompt | Label Names |
|------|--------|-------------|
| **RTE** | His family has steadfastly denied the charges.
question: The charges were denied by his family. True or False?

answer: True | True, False |
| | The charges were denied by his family?
His family has steadfastly denied the charges.

Therefore, the answer is yes. | yes, no |
| | "The charges were denied by his family"?
"His family has steadfastly denied the charges.", so the answer is yes. | yes, no |
| **CB** | He'd gone. Philip had to get them back. His Dad would kill him if he found that he'd taken them.
question: Philip had taken them. true, false, or neither?

answer: true | true, false, neither |
| | Philip had taken them?
He'd gone. Philip had to get them back. His Dad would kill him if he found that he'd taken them.
Therefore, the answer is yes. | yes, no, maybe |
| | "Philip had taken them"?
"He'd gone. Philip had to get them back. His Dad would kill him if he found that he'd taken them."
Therefore, the answer is yes. | yes, no, maybe |
| **WiC** | Room and board.
He nailed boards across the windows.
question: Is the word 'board' used in the same way in the two sentences above?
answer: no | no, yes |
| | "Room and board." / "He nailed boards across the windows.". Similar sense of "board"? No. | No, Yes |
| | Room and board. He nailed boards across the windows. Does "board" have the same meaning in both sentences? No. | No, Yes |
| | board.
- "Room and board." (Sense 1a)
- "He nailed boards across the windows." (Sense 2a) | 2a, 1b |

Table 2: The different prompts we use for RTE, CB, and WiC. We underline the token to predict. For each dataset, the first prompt is the one from GPT-3 [2] and the others are from [9], modified to be compatible with left-to-right LMs when necessary.

Comprehension with Commonsense Reasoning Dataset, 94] is a multiple-choice question-answering task, where each example consists of a news article and a cloze-style question about the article in which one entity is masked out. A system must predict the masked out entity from a list of possible entities in the provided passage.

**Prompts Used** In Table 2, we show the prompts we used for RTE, CB, and WiC in §3.7. Following [3], we also vary the textual label names used to get the logits for a given output class. I.e., for RTE, we use the logit for the word "True" as the probability for the "entailed" class and "False" for the "not entailed" class. We compute class probabilities using a softmax over the above class logits.

## E.3 Dataset and model licenses

LAMA is licensed under CC 4.0.[6] The licenses for SuperGLUE datasets allow for their use and redistribution in a research context (see each individual dataset papers for license details). These datasets do not contain private, personally identifiable information but may contain offensive content. GPT-2/DistilGPT-2 models are licensed under a modified MIT license.[7] GPT-3 models are licensed by OpenAI API to customers via a non-exclusive, non-sublicensable, non-transferable, non-assignable, revocable license.[8]

## E.4 Computing MDL with ADAPET

For MDL as formulated in §2.2, it is not possible to evaluate on the first fold of training data, since the learning algorithm (here, finetuning) requires some initial training data. MDL requires evaluating the loss of the learning algorithm $\mathcal{A}$ on the first fold of data without any training data. Since finetuning is not possible without training data, we say that, in this case, $\mathcal{A}$ returns a uniform distribution over all labels, following prior work [e.g., 32].[9] We use 16 examples (one mini-batch) in the first fold and 2 examples per fold for a remaining 8 folds, to match the number of models we train for CV. As before, we use NLL as the loss $\mathcal{L}$, which is straightforward for most tasks. For WSC and ReCoRD, ADAPET returns class probabilities $\in \{0, 1\}$ which we smooth as $\{\epsilon, 1 - \epsilon\}$ with $\epsilon = 10^{-6}$ to avoid $\infty$ loss values for CV/MDL. For MultiRC, ADAPET makes several binary predictions per example, so we sum the NLLs for these predictions to compute per-example loss.

## E.5 Computational Cost

We use the OpenAI API to evaluate GPT-3 models, costing a total of $2826.73 for all experiments. For GPT-2 experiments, we use a single AMD MI50 GPU (32GB GPU memory) to perform model inference, which requires at most 8 hours (usually less) for all GPT-2/DistilGPT-2 models to evaluate $\mathbb{E}_{R,F}[\text{CV}(\mathcal{A}, R, F)]$, $\mathbb{E}_{R,F}[\text{MDL}(\mathcal{A}, R, F)]$, and expected test accuracy for LAMA and SuperGLUE (any number of training examples). For ADAPET experiments, we use a single AMD MI50 GPU for up to 12 hours to run training and inference for a single model and hyperparamater setting.

---

[6]https://github.com/facebookresearch/LAMA/blob/master/LICENSE

[7]https://github.com/openai/gpt-2/blob/master/LICENSE

[8]https://beta.openai.com/policies/terms-of-use

[9]This technique can be viewed as evaluating the labels' MDL or compression rate where the first fold is compressed using a uniform distribution rather than a learning algorithm.